# Polymer Networks Synthesized from Poly(Sorbitol Adipate) and Functionalized Poly(Ethylene Glycol)

**DOI:** 10.3390/gels7010022

**Published:** 2021-02-20

**Authors:** Haroon Rashid, Yury Golitsyn, Muhammad Humayun Bilal, Karsten Mäder, Detlef Reichert, Jörg Kressler

**Affiliations:** 1Department of Chemistry, Martin Luther University Halle-Wittenberg, D-06120 Halle (Saale), Germany; haroon.rashid@student.uni-halle.de (H.R.); muhammad.bilal@chemie.uni-halle.de (M.H.B.); 2Institute of Pharmacy, Martin Luther University Halle-Wittenberg, D-06120 Halle (Saale), Germany; karsten.maeder@pharmazie.uni-halle.de; 3Department of Physics, Martin Luther University Halle-Wittenberg, D-06120 Halle (Saale), Germany; yury.golitsyn@physik.uni-halle.de (Y.G.); detlef.reichert@physik.uni-halle.de (D.R.)

**Keywords:** networks, hydrogel, crosslinking, enzymatic synthesis, sugar alcohol, functional polyester, Steglich esterification, poly(sorbitol adipate), PEG, solid-state NMR

## Abstract

Polymer networks were prepared by Steglich esterification using poly(sorbitol adipate) (PSA) and poly(sorbitol adipate)-*graft*-poly(ethylene glycol) mono methyl ether (PSA-*g*-mPEG_12_) copolymer. Utilizing multi-hydroxyl functionalities of PSA, poly(ethylene glycol) (PEG) was first grafted onto a PSA backbone. Then the cross-linking of PSA or PSA-*g*-mPEG_12_ was carried out with disuccinyl PEG of different molar masses (Suc-PEG_n_-Suc). Polymers were characterized through nuclear magnetic resonance (NMR) spectroscopy, gel permeation chromatography (GPC), and differential scanning calorimetry (DSC). The degree of swelling of networks was investigated through water (D_2_O) uptake studies, while for detailed examination of their structural dynamics, networks were studied using ^13^C magic angle spinning NMR (^13^C MAS NMR) spectroscopy, ^1^H double quantum NMR (^1^H DQ NMR) spectroscopy, and ^1^H pulsed field gradient NMR (^1^H PFG NMR) spectroscopy. These solid state NMR results revealed that the networks were composed of a two component structure, having different dipolar coupling constants. The diffusion of solvent molecules depended on the degree of swelling that was imparted to the network by the varying chain length of the PEG based cross-linking agent.

## 1. Introduction

Polymer networks are three-dimensional cross-linked structures being applied in various fields, ranging from rubber [1,2] and food industry [3], to bio-medical [4] and pharmaceutical industry [5]. Such polymer architectures are held together by cross-links or junctions, which are formed by strong physical interactions or covalent bonds [6]. The obtained materials possess special properties, such as high elasticity, porosity, swellability etc., which can be modulated to the desired properties of the intended application [3]. Hydrogels or hydrophilic polymer networks are especially well-known for their biomedical applications, because of similar mechanical properties to natural tissues and cells [5,7,8,9].

Networks can be synthesized and modulated according to various factors, like biocompatibility, mechanical strength, hydrophilicity, hydrophobicity, etc. [10,11]. Aliphatic polyesters belong to a class of polymers that have been extensively studied as precursors for the formation of networks, especially for biomedical applications [12,13,14]. Unfortunately, they usually possess a strong hydrophobic character and lack pendant functional groups for conjugation with polymers and biologically active species such as proteins, drugs, or antibodies, etc. [14,15]. Thus, the absence of pendant functional groups along the polymer backbone restricts the modulation for desired material properties [11,16]. Alternatively, enzymatic polymerization can be used for the synthesis of functional polyesters with different advantages [17,18]. It has emerged as a versatile technique for green polymer synthesis over more than two decades, and where enzymes work as biocatalysts to drive the chemical reaction [19,20,21,22]. Thus, it avoids the risk of toxicity which can occur due to metal based catalysts (zinc, aluminum, tin, antimony etc.), used in conventional polycondensation procedures for the production of aliphatic polyesters [22]. Enzymes also render the regioselectivity to the polymerization mechanism under mild reaction conditions producing linear polymers with a low degree of branching, even when multiple OH-group (>2) bearing monomers are employed, such as glycerol or different sugars [23]. It thus can form highly efficient polyesters with pendant functionalities, and avoiding the protection/deprotection chemistry which generally is utilized to functionalize conventional aliphatic polyesters [20,24,25,26]. These polyesters have been utilized in developing different drug delivery carriers, like nanoparticulate systems (nanospheres, nanocubes) [27,28,29,30], micelles [31], microparticles [32], and polymer drug conjugates [29,33,34]. 

Here, we employed enzymatically synthesized poly(sorbitol adipate) (PSA). Every sorbitol part of the linear polyester repeat-unit has four pendant secondary OH-groups, which render the polyester hydrophilic [35]. In order to increase the hydrophilicity of PSA and later the swellability of the networks, some poly(ethylene glycol) (PEG) side-chains were grafted. Finally, using bifunctional PEG with different molar masses, networks with different application profiles were produced and extensively characterized. Various solid-state NMR techniques were employed to characterize the dynamics of the hydrogels swollen in D_2_O.

## 2. Results and Discussion

### 2.1. Polymer Syntheses

Poly(sorbitol adipate) (PSA) was synthesized through lipase CAL-B catalyzed polycondensation reaction using the sugar alcohol sorbitol and divinyl adipate. The selection of divinyl esters was preferred over dicarboxylic acids or dialkyl esters because the latter result in reaction by-products like water, methanol, etc. which require vacuum and high temperatures in order to be removed from the reaction mixture [36]. Another reason to select divinyl adipate was connected with the fact that relatively high molar mass polyesters are obtained. Here, the byproduct vinyl alcohol, which tautomerizes spontaneously to acetaldehyde, leaves the reaction as a gas. This process drives the reaction irreversibly in a forward direction [37,38,39]. The sugar alcohol sorbitol was selected since the two primary OH-groups are converted during the polycondensation process, but four secondary OH-groups per repeat unit of the polyester remain free, which guarantee the water solubility of PSA [40]. Hence, a linear polyester is achieved due to the selectivity of CAL-B towards primary hydroxyl groups rather than secondary ones [41]. The PSA structure was confirmed from ^1^H NMR, ^13^C NMR, and ^13^C MAS NMR spectra given in Figure 1a, Appendix A, respectively.

Our objective was to synthesize biocompatible PSA networks which could be used as hydrogel for potential applications in pharmacy and medicine. For this purpose, PSA was first modified by grafting with PEG chains. This further improves the hydrophilicity of the polymer. PEG is known as a versatile biocompatible polymer with a well-recognized safety profile. For these reasons, it has been in use for decades for different purposes in the pharmaceutical industry [42]. To achieve the grafting reaction, mPEG_12_-OH was first reacted with succinic anhydride in the presence of 4-(dimethylamino)pyridine (DMAP) to obtain the monofunctional α-methoxy,ω-succinyl poly(ethylene glycol) (mPEG_12_-Suc) (Appendix A). The reaction was verified by ^1^H NMR spectroscopy (Appendix A).

To synthesize PSA-*g*-mPEG_12_, Steglich esterification was adopted for the grafting procedure via the reaction between free hydroxyl groups from PSA and carboxyl groups from mPEG_12_-Suc. The synthesis of PSA-*g*-mPEG_12_ could be confirmed from ^1^H NMR (Figure 1b) and ^13^C NMR spectra (Figure 2). The appearance of the methyl peak at 3.23 ppm and methylene peak at 3.50 ppm of PEG in the ^1^H NMR spectrum, while in the ^13^C NMR spectrum, the carbon signal of the methyl group at 58.10 ppm and the carbon signal of the methylene group at 69.85 ppm of PEG, verified the grafting of the mPEG_12_ chain to PSA. Equation (1) was employed for the calculation of the degree of grafting in mol%.
(1)$$\mathrm{Degree}\;\mathrm{of}\;\mathrm{grafting}\;=\frac{1/3\int d}{1/4\int a\;}\;\times \;100$$
where *d* represents the protons of the methyl group of PEG, and *a* represents the protons of the methylene groups of PSA.

The degree of grafting was calculated as ~45 mol% per polymer chain, i.e., nearly one out of eight OH-groups is esterified with mPEG_12_-Suc. The degree of grafting can also be confirmed by taking the same ^1^H NMR integrals of the mentioned peaks with CDCl_3_ as solvent, given in Appendix A.

The grafting was also verified by gel permeation chromatography (GPC) traces through a shift towards shorter retention time (Appendix A). Suc-PEG_n_-Suc was synthesized through carboxylation of OH-PEG-OH on both sides with succinic anhydride. By the same procedure as followed for the synthesis of mPEG_12_-Suc, esterification took place between OH-PEG-OH and succinic anhydride. Using Suc-PEG_n_-Suc, with varying chain lengths of PEG, polymer networks were then prepared through Steglich esterification. Esterification took place between hydroxyl groups from PSA or PSA-*g*-mPEG_12_ and the carboxylate groups of the succinyl part of Suc-PEG_n_-Suc in order to study the effect of different chain lengths of the cross-linkers on the overall behavior of the networks. An ideal network structure is given in Figure 3, while it can be verified by ^13^C CP MAS NMR spectra (Appendix A).

### 2.2. Differential Scanning Calorimetry

Figure 4 and Table 1 summarize all DSC data taken in the range between −60 °C and 80 °C. Figure 4a indicates the amorphous nature of PSA and PSA-*g*-mPEG_12_, since only a glass transition temperature (*T_g_*) is observed in the respective heating trace. In contrast, mPEG_12_-Suc shows a clear melting endotherm. Thus, the amorphous nature of PSA-*g*-mPEG_12_ indicates that the PSA backbone prevents the crystallization of the grafted PEG chains. PSA shows a *T_g_* at −1 °C which is reduced to −34 °C after grafting with mPEG_12_-Suc, and resulting in PSA-*g*-mPEG_12_. The bifunctional cross-linker Suc-PEG_9_-Suc also did not show any melting peaks but showed a *T_g_* at −45 °C (see Figure 4b). When PSA was cross-linked with Suc-PEG_9_-Suc, the *T_g_* was lowered to −11 °C. When PSA-*g*-mPEG_12_ was cross-linked with Suc-PEG_9_-Suc, the *T_g_* appeared at −29 °C. Both network structures cross-linked with Suc-PEG_9_-Suc did not show any melting peaks, demonstrating that the networks formed with Suc-PEG_9_-Suc were completely amorphous. In contrast, in networks formed with Suc-PEG_23_-Suc and Suc-PEG_45_-Suc, respectively, cross-linkers were semi-crystalline, indicated by their melting endotherms in the DSC traces, no matter if they were synthesized from PSA or PSA-*g*-mPEG_12_ (Figure 4c,d). Here, the crystallinity is obviously imparted by the cross-linkers Suc-PEG_23_-Suc and Suc-PEG_45_-Suc, since they are also semi-crystalline.

These traces also indicated a decrease in the melting temperature and melting enthalpies when the cross-linker molecules were located between PSA-based polymer backbones compared to the native cross-linker. For the PSA networks formed with Suc-PEG_23_-Suc, the phenomenon of cold-crystallization could also be observed [43]. Obviously, all networks formed by Suc-PEG_9_-Suc are amorphous, but when Suc-PEG_23_-Suc is employed for network formation, it needs some thermal activation so that the PEG_23_-chains can form crystals. In the case of Sus-PEG_45_-Suc, the PEG_45_-chains are long enough to be packed into polymer crystals immediately after the synthesis of the networks. Furthermore, Suc-PEG_45_-Suc based networks show higher melting temperatures compared to Suc-PEG_23_-Suc based networks [44,45].

### 2.3. Swelling Studies

Swelling studies of the networks were performed in D_2_O up to maximum water uptake (see Figure 5). This point can be considered as the equilibrium degree of swelling [11,46]. The studies revealed that the degree of swelling of all networks increased with the chain length of PEG based cross-linkers. There exists a nearly linear relation between the equilibrium degree of swelling and the degree of polymerization *n* of the PEG-based bifunctional cross-linkers. Both PSA and PSA-g-mPEG_12_ networks showed the highest degree of swelling when Suc-PEG_45_-Suc was used as cross-linker. In comparison to PSA networks, the PSA-g-mPEG_12_ networks showed higher degrees of swelling, since the grafted PEG chains rendered the networks more hydrophilic. Figure 6 shows photographic images as examples of the different water uptake at room temperature. The images verify the two most important aspects of the swelling experiments: (i) The degree of swelling was larger for PSA-g-mPEG_12_ samples compared to PSA based samples using identical cross-linkers; and (ii) The degree of swelling increased with the degree of polymerization *n* of the cross-linker molecules when identical precursor polymers were employed, i.e., PSA and PSA-g-mPEG_12_, respectively.

### 2.4. ^1^H Double-Quantum (DQ) NMR Spectroscopy

For the characterization of the network samples, ^1^H DQ measurements were performed. This method allows the analysis of the dynamics and the determination of the network structure. It also permits distinguishing the network-forming loops of different lengths, as well as various defects [47]. The exact details of the double-quantum (DQ) experiment and the pulse sequence according to Baum and Pines [48] used in this work can be taken from Ref. [49].

For the determination of the dipolar coupling constant, the mobile and uncoupled components, the so-called *tail* fraction, are first subtracted from the signal. Two-*tail* components are expected in a swollen real polymer network. The first is the solvent *tail*, which is caused by residual protons in the solvent, and since the samples contain an excess of D_2_O, an increased amount of this component is expected. Second is the network *tail*, which is caused by the network defects, such as loops and dangling chain ends (unreacted chains).

For unmodified samples of PSA networks (Appendix A), the solvent *tail* could only be detected. Based on the long *T*_2_*** decay time, it can be excluded that this component resulted from the network. The free chain ends of the main strands could not be detected. Note that the solvent *tail* did not correspond to the total amount of water in the sample. The DQ experiments were performed with a repetition time of 2 s. This delay was chosen to be sufficient for the complete relaxation of the polymer, but was significantly shorter than 5·*T*_1_ of the solvent. Thus, the short recycling delay acted as a *T*_1_ filter for the solvent, and the determined water content was lower than expected.

Even in modified networks of PSA-*g*-mPEG_12_, where about ~45% of the monomer units contained a side chain, no second *tail* component with a short *T*_2_* time could be detected (Appendix A). This result is surprising and indicates that the network *tail*, i.e., the free chain ends and other defects, did not differ in their mobility from the network structure. In other words, the isotropic mobility of the tail is restricted by the network.

To check the *tail* subtraction procedure, the normalized DQ intensity (*I_nDQ_*) was calculated (Appendix A). The *nDQ* intensity reaches the expected value of 0.5 for all samples, which is an indication that the correct tail fraction has been subtracted. It should be noted, however, that due to a complex network structure, the DQ data may contain not just one (as is the case of natural rubber), but several network components with different *T*_2_* values. This makes the *nDQ* curves unsuitable for a quantitative evaluation, due to the incorrect normalization.

For this reason, the DQ data was directly evaluated with the help of a simultaneous fitting procedure. The PEG spacers as cross-linkers, which connect the main chains, are generally randomly distributed. This means that there are no loops of well-defined length like in Tetra-PEG [47] or other PEG networks [50], and this was the case in these samples. The mesh size can vary greatly, which leads to a broad distribution of dipolar couplings due to the relationship *D_res_* ∝ *M_c_*, the average molar mass between the cross-links. We tested both the Gaussian and log-normal distribution, and the latter showed a better fit to the measured data. The simultaneous fitting function for two network components is
(2)$$I_{\sum DQ}\left(\tau_{DQ}\right)\;=\overset{2}{\underset{i=1}{\sum }}f_{i}exp\left\{-{(\tau_{DQ}/\tau_{i})}^{\beta_{i}}\right\}$$
(3)$$I_{DQ}\left(\tau_{DQ}\right)\;=\overset{2}{\underset{i=1}{\sum }}\int_{0}^{\infty }I_{DQ}\left(\tau_{DQ},\;D_{res}^{\left(i\right)}\right)p\left(D_{res}^{\left(i\right)}\;,\sigma^{\left(i\right)}\right)dD_{res}^{\left(i\right)}$$
(4)$$I_{DQ}\left(\tau_{DQ}\right)\;=\overset{2}{\underset{i=1}{\sum }}\int_{0}^{\infty }\frac{1}{2}f_{i}\left[1-exp\left\{-{\left(0.378\cdot D_{res}^{\left(i\right)}\tau_{DQ}\right)}^{1.5}\right\}\cdot cos\left(0.583\cdot D_{res}^{\left(i\right)}\tau_{DQ}\right)\right]\cdot \exp\left\{-{\left(\frac{\tau_{DQ}}{\tau_{i}}\right)}^{\beta_{i}}\right\}\cdot p\left(D_{res}^{\left(i\right)}\;,\sigma^{\left(i\right)}\right)dD_{res}^{\left(i\right)}$$

Fitting parameters are the component fraction (*f_i_*), the time constant (*τ_i_*), which corresponds to the characteristic decay time (*T*_2_*), the exponent *β* of the Kohlrausch–Williams–Watts (KWW) function, the residual dipolar coupling constant *D_res_*, and the width of the log-normal distribution *σ*.

Table 2 summarizes the fitting results. The most important parameters are the component fractions (*f_i_*), the residual dipolar coupling constants (*D_re_*_s_), and the distribution widths (*σ*). The corresponding fitting curves are shown in Appendix A. Two network components with significantly different coupling constants were detected for all samples. The weakly coupled component dominated for all samples. With increasing length of the PEG spacers, the network became more mobile (*T*_2_* value increases and *D_res_* decreases) and the proportion of more strongly coupled spins decreased. This applied generally to both PSA and PSA-*g*-mPEG_12_ network batches.

The difference between PSA and PSA-*g*-mPEG_12_ networks lies in the strength of the dipole coupling. As can be seen from Figure 7, PSA networks are more strongly coupled. However, the proportion of such strongly coupling protons is lower in PSA-*g*-mPEG_12_ networks. In addition, there is a greater distribution width of the dipole couplings. The additional side chains thus lead to increasing inhomogeneity in the network. Not to be excluded is the case of phase separation, with areas of high and low polymer concentration.

If the side chains and free chain ends are located in the first areas, they are less mobile and show a residual dipole coupling. This could explain the “invisible” tail in the PSA-*g*-mPEG_12_ network samples. Keeping in view data from Table 2 and Figure 7, dipolar coupling was decreasing, while *T*_2_* was increasing, causing mobility of the network. In other words, the increase in network mobility can be attributed to an increase of the degree of swelling, as shown in Figure 7.

### 2.5. ^1^H Pulsed Field Gradient (PFG) NMR Spectroscopy

Furthermore, the diffusion of solvent molecules in swollen networks was investigated (See Figure 8). The samples were prepared without excess D_2_O, so that diffusion outside the polymer network could be excluded. The measurement signal corresponded exclusively to the remaining HDO molecules in the polymer network.

The ^1^H spectra from the diffusion measurements are shown in Appendix A, and exhibit two well-resolved resonances. As the gradient strength (*G*) increases, the intensity of the left peak decreases while the right peak remains constant. The assignment of the two resonances is therefore obvious. The left peak corresponds to the mobile solvent (HDO), while the right peak is assigned to the immobile polymer network (its diffusion can be neglected on the time scale of the diffusion of water). The relationship between the echo signal intensity and pulse field gradient parameters in the PFG experiment is given by
(5)$$I_{PFG}\left(G\right)=I_{PFG}\left(0\right)\cdot exp\left(-\gamma^{2}G^{2}D\delta^{2}\left(\Delta -\delta /3\right)\right)$$
where *I_PFG_* is the echo signal intensity, *γ* is the gyromagnetic ratio of the proton, *G* is the field gradient strength during the gradient pulse of the length (*δ*), *D* is the self-diffusion coefficient, and Δ is the diffusion time [51].

The echo signal intensity was measured as a function of *G*. When plotting the logarithm of the echo intensity against the diffusion function *γ*^2^*G*^2^*δ*^2^(Δ-*δ*/3), the diffusion coefficient (*D*) can be determined from the slope of the curve. 

In order to determine the diffusion coefficients of water, only the HDO peak was integrated. Figure 8 shows the normalized PFG diffusion decays of HDO of the investigated samples. In contrast to the diffusion of HDO in pure D_2_O, the decay curves of the network samples showed a clear nonlinear behavior. This nonlinearity can be explained in two different ways. On one hand, it could be an anisotropic diffusion movement of water, e.g., caused by hydrogen bonding with the polymer. In this case, the evaluation of the diffusion data would be more difficult. However, if the diffusion length of an HDO molecule is calculated on the time scale of the diffusion measurement, it becomes clear that it is significantly larger than the dimensions of the smallest possible loop in the polymer network. The anisotropy of diffusion can thus be excluded. On the other hand, the non-linearity of the diffusion curves could be caused by a baseline problem in the spectrum. When looking closely at the spectra in Appendix A, it becomes clear that the HDO peak overlaps with the network peak. This results in an offset in the diffusion curve, which must be taken into account. In this case, the upper formula is revised with a constant intensity offset. It is not necessary to add a second diffusion component, because the diffusion movement of the polymer network can be neglected on the time scale of the measurement. The strength of the background signal and thus the amount of offset depends on the spectral resolution and on the *T*_2_ relaxation time of the polymer, and varies from sample to sample.

The determined diffusion coefficients are shown in Figure 9 as a function of the degree of swelling *Q*, and are in good agreement with the data reported for the poly-(*N*,*N*-dimethylacrylamide) gels [52]. Especially at low swelling degrees, the network environment significantly influenced the normal diffusion movement of water. With increasing length of the PEG spacers, water diffusion in the polymer network became faster and approached the value of pure D_2_O. In a direct comparison of the two network batches, it can be seen that the PSA-*g*-mPEG_12_ networks could absorb more water, and therefore always showed higher diffusion coefficients for the same length of PEG spacers.

## 3. Conclusions

Enzymatic polymerization has been used as an alternative approach to produce green aliphatic polyesters, avoiding the drawbacks associated with conventional polymerization techniques. The presence of multiple hydroxyl groups on the PSA backbone not only allowed us to modify it with PEG side chains, but also to use these functional groups for the synthesis of polyester-based networks. All syntheses of PSA, mPEG_12_-Suc, PSA-*g*-mPEG_12_, and Suc-PEG_n_-Suc (bifunctional cross-linker) were confirmed through solution NMR spectroscopy. DSC data indicated the amorphous nature of PSA and PSA-*g*-mPEG_12_, and its influence on the melting temperatures of PEG segments in the networks incorporated by semi-crystalline cross-linker molecules. This shows that networks formed with low molar mass PEG cross-linkers (Suc-PEG_9_-Suc) are amorphous, while networks with high molar mass PEG cross-linkers (Suc-PEG_23_-Suc and Suc-PEG_45_-Suc) are semi-crystalline due to the crystallinity of the PEG based precursors. Furthermore, the degree of swelling of the network samples was directly related to the chain length of cross-linking agents. The structure and dynamics in PSA polymer networks were investigated by ^13^C MAS and ^1^H DQ NMR measurements. Experimental data indicated that the networks had an inhomogeneous structure. For both PSA and PSA-*g*-mPEG_12_ networks, two network components with different dipolar coupling constants were detected. The exact amount of network defects could not be determined due to the inhomogeneity of the networks. ^1^H PFG diffusion measurements on swollen networks allowed the determination of the diffusion coefficients of HDO in the networks. As expected, the diffusion of the solvent strongly depended on the degree of swelling of the sample. The modification of the PSA samples by PEG side chains led to increased swelling, and thus to faster diffusion in the network. Keeping in view the tunable characteristics of the networks as an exciting drug delivery system in general, as well as current physico-chemical data and the polymerization process in specific, potential pharmaceutical applications seem possible.

## 4. Materials and Methods

### 4.1. Materials

Novozyme 435, Lipase derived from *Candida Antarctica* type B (CAL-B) and immobilized on acrylic resin, was purchased from Sigma Aldrich, St. Louis, MO, USA. It was vacuum dried over phosphorous pentoxide for 24 h prior to use. Sorbitol (98%) and divinyl adipate were purchased from Sigma Aldrich (Steinheim, Germany) and TCI GmbH (Eschborn, Germany), respectively. Phosphorous pentoxide (≥99%), 4-(dimethylamino)pyridine (DMAP), anhydrous *N*,*N*-dimethylformamide (DMF, 99.8%), anhydrous tetrahydrofuran (THF, 99.9%), 1-ethyl-3-(3-dimethylaminopropyl)carbodiimide hydrochloride (EDC·HCl), dialysis membranes with 1000 g·mol^−1^ molar mass cut off (MWCO) and 10,000 g·mol^−1^ MWCO (Spectra/Por^®^, made from regenerated cellulose) were purchased from Carl Roth, Karlsruhe, Germany. Deuterated chloroform (CDCl_3_) and deuterated dimethyl sulfoxide (DMSO-d_6_) were purchased from Armar (Europa) GmbH (Leipzig, Germany). α,ω-bis-hydroxy poly(ethylene glycol)_n_ (OH-PEG_n_-OH, with n = 9, 23, and 45) and α-methoxy,ω-hydroxy poly(ethylene glycol)_12_ (mPEG_12_-OH) were purchased from Alfa Aesar (Kandel, Germany).

### 4.2. Synthesis of Poly(Sorbitol Adipate) (PSA)

PSA was synthesized by enzymatic polymerization, as described elsewhere [40]. In short, an equimolar amount of sorbitol (10.0 g, 54.9 mmol) and divinyl adipate (DVA) (10.88 g, 54.9 mmol) was added to a 250 mL three neck round bottom flask. The flask was connected with a reflux condenser with a calcium chloride drying tube and to a mechanical stirrer. It was then charged with 50 mL acetonitrile and stirred for 30 min at 50 °C until the temperature was equilibrated. Novozyme 435 (2.1 g, 10% *w*/*w* of total mass of PSA and DVA) was then added to start the polymerization. The reaction mixture was stirred for 92 h, and was then stopped and diluted with DMF followed by the removal of enzyme beads by filtration with Whatman^®^ filter paper. The concentrated filtrate was processed through dialysis against deionized water for 7 days using a dialysis membrane with 1000 MWCO. Finally, the polymer solution was freeze dried to obtain the final pure product of PSA. The purity of the product was confirmed from ^1^H NMR spectroscopy (Figure 1a). ^1^H NMR (400 MHz, DMSO-d_6_) *δ* (ppm): 4.95–4.58 (m, 2H), 4.57–4.33 (m, 2H), 4.28–3.86 (m, 2H), 3.82–3.72 (m, 2H), 3.61–3.34 (m, 2H), 2.38–2.18 (m, 4H), and 1.61–1.41 (m, 4H).

### 4.3. Synthesis of Mono- and Bifunctional PEG

In a typical procedure to synthesize α,ω-bis-succinyl poly(ethylene glycol) (Suc-PEG_n_-Suc) (with n = 9, 23, 45) and α-methoxy,ω-succinyl poly(ethylene glycol)_12_ (mPEG_12_-Suc), PEG was acylated by reaction with succinic anhydride through a procedure described elsewhere [53,54]. For the synthesis of mPEG_12_-Suc, mPEG_12_ of molar mass 550 g·mol^−1^ was used, while for the synthesis of Suc-PEG_n_-Suc, OH-PEG_n_-OH having molar mass of 400 g·mol^−1^, 1000 g·mol^−1^, and 2000 g·mol^−1^ were used, respectively. Suc-PEG_n_-Suc ^1^H NMR ((400 MHz, CDCl_3_) *δ* (ppm), Appendix A): 4.28–4.20 (m, 4H), 3.73–3.57 ((m, 34H (Suc-PEG_9_-Suc); 92H (Suc-PEG_23_-Suc); 180H (Suc-PEG_45_-Suc)), 2.68–2.58 (m, 8H). mPEG_12_-Suc ^1^H-NMR ((400 MHz, CDCl_3_) *δ* (ppm), Appendix A): 4.25–4.21 (m, 2H), 3.68–3.51 (m, 50H), 3.35 (s, 3H), 2.67–2.56 (m, 4H).

### 4.4. Synthesis of PSA-g-mPEG_12_

PEG was introduced as a side chain to PSA through Steglich esterification, by reacting pendant hydroxyl groups of PSA and carboxyl groups from mPEG_12_-Suc. The overall procedure was as follows: PSA (2.0 g, 27.3 mmol) and mPEG_12_-Suc (2.67g, 4.11 mmol) were charged into a two neck round bottom flask together with anhydrous DMSO. DMAP (0.15 g, 1.2 mmol) and EDC·HCl (2.35 g, 12.3 mmol) were added as catalysts to the reaction mixture. It was stirred for 24 h at room temperature. The crude product was then purified by dialyzing it in deionized water through a dialysis membrane for 5 days, using a membrane with MWCO 10,000 g·mol^−1^. The diluted product solution was then freeze dried to obtain the final product. ^1^H NMR ((400 MHz, DMSO-d_6_) *δ* (ppm) (Figure 1b)): 4.95–4.58 (m, 2H), 4.57–4.33 (m, 2H), 4.16–4.08 (m, 2H), 4.28–3.86 (m, 3H), 3.82–3.72 (m, 2H), 3.56–3.45 (m, 50H), 3.23 (s, 3H), 2.61–2.52 (m, 4H), 3.61–3.34 (m, 2H), 2.38–2.18 (m, 4H), 1.61–1.41 (m, 4H).

### 4.5. Network Syntheses of PSA or PSA-g-mPEG_12_ with Suc-PEG_n_-Suc

Networks were synthesized by esterifying free hydroxyl groups from PSA or PSA-*g*-mPEG_12_ with carboxyl groups of Suc-PEG_n_-Suc (with n = 9, 23, 45) using Steglich esterification as shown in Scheme 1. In a typical experiment, PSA (0.500 g, 6.85 mmol) was first dissolved in DMSO followed by the addition of DMAP (0.17 g, 1.36 mmol) and EDC·HCl (2.61 g, 13.69 mmol) in a vial. At the end, Suc-PEG_9_-Suc (1.23 g, 2.05 mmol) was added to the above reaction mixture and kept overnight at 37 °C to obtain PSA gels.

A similar procedure was adopted to synthesize networks from PSA-*g*-mPEG_12_. For both PSA and PSA-*g*-mPEG_12_ based networks, three different Suc-PEG_n_-Suc cross-linkers with varying chain lengths were used (with n = 9, 23, 45). The same concentration, i.e., 30 mol% of cross-linker with respect to free hydroxyl groups available at the polymer backbone, was used for the syntheses of both types of network. The gels were then cut into discs and washed with deionized water as washing medium, with repeated replacements 3 times per day. The washing process was continued for 7 days in order to remove all impurities present in the networks. Purified swollen network discs were then dried in a vacuum oven at 37 °C to obtain the dry and clean product. The synthesis scheme for polymer precursors and network formation are also given in Scheme 1.

### 4.6. Nuclear Magnetic Resonance (NMR) Spectroscopy

#### 4.6.1. Solution NMR Spectroscopy

Solution NMR spectroscopy (^1^H NMR and ^13^C NMR) was performed using an Agilent VNMRS spectrometer 400 MHz at 27 °C. Tetramethylsilane (TMS) was used as internal calibration standard. Deuterated solvents, DMSO-d_6_ and CDCl_3_, were used for measuring the spectra of the polymers. Measurements were evaluated through MestReNova software (version 11.0.4), developed by Mestrelab Research, Spain, while the peaks were assigned using ChemDraw Ultra software (version 7.0), developed by CambridgeSoft Corporation, Cambridge, MA, USA.

#### 4.6.2. Solid State NMR Spectroscopy

##### ^13^C Magic Angle Spinning (MAS) NMR Spectroscopy

^13^C MAS cross-polarization (CP) and single-pulse (SP) experiments were performed on a Bruker Avance 400 spectrometer, operating at a ^13^C Larmor frequency of 100 MHz. A 4 mm triple-resonance probe head was used and a MAS spinning frequency of 10 kHz was applied in all experiments. The sample temperature was controlled by a standard Bruker VT-controller and calibrated with methanol. For all experiments, the temperature was set to 30 °C. The ^13^C π/2 pulse length varied between 2.5 and 3.0 μs. The recycle delay was chosen so as to meet the condition of 5·*T*_1_ of protons, to allow complete restoration of the initial signal. The corresponding relaxation times were estimated by means of the saturation–recovery pulse sequence. Dry samples were filled into a Kel-F insert for a standard 4 mm rotor. The spectrum was referenced according to the COO resonance of alanine. The line assignment was carried out using ChemDraw Ultra software (version 7.0) developed by CambridgeSoft Corporation, USA.

##### ^1^H Double Quantum (DQ) NMR Spectroscopy

^1^H DQ NMR experiments were performed on a Bruker Avance III 200 MHz spectrometer using a static 5 mm Bruker probe. The temperature was controlled with a BVT-3000 heating-device with an accuracy of ±1 °C. For all experiments, the temperature was set to 30 °C. ^1^H π/2 pulse of 3 μs length and recycle delay of 2 s were applied. For the DQ measurements, samples were swollen in D_2_O to equilibrium and then filled into 5 mm glass tubes. A small amount of D_2_O was added to prevent the samples from drying out during the long measurement. Afterwards the tube was sealed.

##### ^1^H Pulsed Field Gradient (PFG) NMR Spectroscopy

^1^H PFG NMR spectroscopy was carried out for diffusion coefficient measurements. The spectra were recorded with a Bruker Avance II 400 MHz instrument at 30 °C. A stimulated echo with bipolar gradient (STEBP) was used as sequence with gradient time *δ* of 1–2 ms and a varying diffusion time Δ of 20–80 ms, depending on the sample. ^1^H π/2 pulse length between 3.0 and 3.5 μs and recycle delays of at least 5·*T*_1_ were applied. Samples swollen in D_2_O to equilibrium were filled into 5 mm glass tubes. The amount of HDO molecules in deuterated water is sufficient to obtain a good signal, which is comparable in intensity to the signal from the network. No additional D_2_O was added. This ensures that there was no signal contribution from free water outside the network. Since the diffusion measurements usually take a few hours, and were performed at room temperature, the drying of the samples could be neglected.

### 4.7. Gel Permeation Chromatography (GPC)

GPC measurements were performed using a Viscotek GPCmax VE 2002 having columns of HHRH Guard-17360 and GMHHR-N-18055 with refractive index detector (VE 3580 RI detector, Viscotek) at room temperature. DMF with the addition of 0.01 M LiBr was used as an eluent with a sample concentration of 5 mg·mL^−1^. Poly(methyl methacrylate) (PMMA) was used as calibration standard and the flow rate was 1 mL·min^−1^. The number of average molar mass (*M_n_*), the weight average molar mass (*M_w_*), and the dispersity (*Ð*, *M_w_/M_n_*) were determined.

### 4.8. Differential Scanning Calorimetry (DSC)

The thermal analysis of all precursor polymers and networks was carried out using a DSC, Mettler Toledo DSC823e module, Mettler-Toledo GmbH, Greifensee, Switzerland. Pre-weighed samples were placed in aluminum crucibles and were scanned against temperature ranging from −60 °C to 80 °C. A heating rate of 1 °C·min^−1^ was employed and the nitrogen flow rate was 10 mL·min^−1^.

### 4.9. Swelling Studies

Dried network discs were investigated for solvent uptake studies in D_2_O. In a typical experiment, pre-weighed dry network samples were immersed into water and allowed to swell for 24 h at room temperature until they reached their equilibrium state. Swollen discs of both PSA and PSA-*g*-mPEG_12_ based networks were then taken out and rolled over blotting paper in order to remove water from the surface and then weighed. The degree of swelling at equilibrium swelling was finally calculated by taking into account the weights of dry and swollen samples. Measurement was done in triplicate. The degree of swelling was calculated using Equation (6).
(6)$$\mathrm{Degree}\;\mathrm{of}\;\mathrm{swelling}\;=\frac{m_{t}-\;m_{o}}{m_{o}}$$
where, *m_t_* refers to the swollen network, while *m_o_* refers to the dried network disc.

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
