# Peer review of "Polymer Networks Synthesized from Poly(Sorbitol Adipate) and Functionalized Poly(Ethylene Glycol)"

_gels, 2021, doi:10.3390/gels7010022_

Round 1
Reviewer 1 Report
The present paper deals with polymer networks prepared by Steglich esterification using poly(sorbitol adipate) (PSA) and poly(sorbitol adipate)-graft-poly(ethylene glycol) mono methyl ether copolymer. The authors utilize the multiple hydroxyl functionalities of the sorbitol fragment to graft PEG onto PSA backbone. The research is well conducted, with a solid and extensive experimental characterization, including the degree of swelling. The paper is therefore well suited for the journal Gels. I recommend the authors include relevant references in the field of crosslinkable, biodegradable polymers:
DOI: 10.1039/C3BM60171D; doi: 10.1002/bit.22361; DOI: 10.1039/D0PY00843E, DOI: 10.1021/bm049492o
Author Response
Reviewer No. 1:
The present paper deals with polymer networks prepared by Steglich esterification using poly(sorbitol adipate) (PSA) and poly(sorbitol adipate)-graft-poly(ethylene glycol) mono methyl ether copolymer. The authors utilize the multiple hydroxyl functionalities of the sorbitol fragment to graft PEG onto PSA backbone. The research is well conducted, with a solid and extensive experimental characterization, including the degree of swelling. The paper is therefore well suited for the journal Gels. I recommend the authors include relevant references in the field of crosslinkable, biodegradable polymers:
DOI: 10.1039/C3BM60171D; doi: 10.1002/bit.22361; DOI: 10.1039/D0PY00843E,
DOI: 10.1021/bm049492o
Response:
Two of the references (reference no.8 and no. 9), which are relevant to our manuscript, have been added. (Page 1, Line 39)
Reviewer 2 Report
The manuscript reports on the synthesis and swelling properties of polyester networks based on enzymatically synthesized poly(sorbitol adipate) (PSA). In order to impart hydrophilicity and hence to increase swelling of the networks in water, PSA macromolecules are first grafted with short PEG-chains and then crosslinked by using bifunctional PEG-reagents of varied chain length.
The work is accurately designed and well elaborated, all synthesized polymers and networks are thoroughly evaluated by applying adequate methods including advanced solid-state NMR techniques. The discussion is comprehensive and conclusions are well supported by experimental results.
In my opinion this manuscript could be accepted for publication in Gels after minor revision.
Comments and suggestions:
- In the manuscript the authors use term polydispersity index / PDI (e.g. page 6, line 226; page 16, line 605). According to IUPAC it is recommended to use term dispersity, represented by the symbol Đ.
- In the caption of Figure 5 it is stated: ”Degree of swelling … in D2O”, whereas on p. 16: “Dried network discs were investigated for solvent uptake studies in deionized water”. Please specify if different solvents (water, heavy water or HDO) are used and in which cases.
- Vertical axis of Figure 5 is named “Degree of Swelling (%)”. Considering the equation (6) on p. 16 the degree of swelling is not measured in %. The axis’ name should be changed.
- In Scheme S1 in the supplementary file, the structure of starting poly(ethylene glycol) methyl ether is wrongly drawn as PEG propyl ether. The scheme should be redrawn.
Author Response
Reviewer No. 2:
The manuscript reports on the synthesis and swelling properties of polyester networks based on enzymatically synthesized poly(sorbitol adipate) (PSA). In order to impart hydrophilicity and hence to increase swelling of the networks in water, PSA macromolecules are first grafted with short PEG-chains and then crosslinked by using bifunctional PEG-reagents of varied chain length.
The work is accurately designed and well elaborated, all synthesized polymers and networks are thoroughly evaluated by applying adequate methods including advanced solid-state NMR techniques. The discussion is comprehensive and conclusions are well supported by experimental results.
In my opinion this manuscript could be accepted for publication in Gels after minor revision.
Comments and suggestions:
- In the manuscript the authors use term polydispersity index /PDI (e.g. page 6, line 226; page 16, line 605). According to IUPAC it is recommended to use term dispersity, represented by the symbol Đ.
Response: IUPAC recommended term dispersity and symbol Đ have been added. (Page 6, Line 227 including Table 1 and Page 16, Line 606)
- In the caption of Figure 5 it is stated: ”Degree of swelling … in D2O”, whereas on p. 16: “Dried network discs were investigated for solvent uptake studies in deionized water”. Please specify if different solvents (water, heavy water or HDO) are used and in which cases.
Response: Swelling measurements were done in D2O. Therefore, D2O has been added to the text. (Page 16, Line 612)
- Vertical axis of Figure 5 is named “Degree of Swelling (%)”. Considering the equation (6) on p. 16 the degree of swelling is not measured in %. The axis’ name should be changed.
Response: % has been removed from the vertical axis of Figure 5.
- In Scheme S1 in the supplementary file, the structure of starting poly(ethylene glycol) methyl ether is wrongly drawn as PEG propyl ether. The scheme should be redrawn.
Response: Scheme has been redrawn. (Scheme S1, supplementary file)
Reviewer 3 Report
The study of the authors on their PEG functionalised Sorbitol-based hydrogels is presented in a very nice manner. In addition to the way it is presented, it is also a rather wholistic study that looks at the prepared hydrogels from a variety of angles and includes some NMR techniques (like Magic Angle Spinning NMR), which should be more common in the field than they currently are. The paper should be published after following the following minor revisions:
P2: The authors state that aliphatic polymers would generally hydrophobic. Looking at PMOXA, Polysarcosin and PAA, amongst others, these statement needs to be reconsidered
P3: It should be mentioned earlier that GPC was done as well. The discussion can be as it is later on, but a brief mentioning would be good.
P4: The characteristic of the PEG-grafted Geld should be pronounced better in the phrasing on this page, but also throughout the manuscript. It seems that the authors sometimes lose track of this important characteristic.
P5-6: PEG 45 melts higher that PEG23, but this is not reflected as such (only the higher degree of crystallinity is mentioned). The melting temperatures have to be cross-referenced with literature data.
P6 (Table): Since PSA-g-mPEG12 allows for an absolute calculation of the molar mass, this should be used be used to back-calculate the original molar mass of PSA to check the value by GPC. GPC values can be off by a margin, so this rather important to check.
P7 - Fig. 5: The measurements appear to be done only once. They have to repeated to be triplicates in order to add error bars and put the results into perspective
P7 - Fig. 6: The figure caption says the gels swell in D20 while the text mentions H20. Please be consistent and correct where necessary.
P11 - Fig. 9: The figure actually contains error bars, but these are too small to be recognised (or are hard to spot). This should be stated in the figure caption.
If these measures are taken, the manuscript can be published. Especially the NMR discussion and the discussion of the translational diffusion coefficient are really well done!
Author Response
Reviewer No. 3:
The study of the authors on their PEG functionalised Sorbitol-based hydrogels is presented in a very nice manner. In addition to the way it is presented, it is also a rather wholistic study that looks at the prepared hydrogels from a variety of angles and includes some NMR techniques (like Magic Angle Spinning NMR), which should be more common in the field than they currently are. The paper should be published after following the following minor revisions:
- P2: The authors state that aliphatic polymers would generally hydrophobic. Looking at PMOXA, Polysarcosin and PAA, amongst others, these statement needs to be reconsidered.
Response: We are specifically referring to the aliphatic polyesters as hydrophobic and not aliphatic polymers in general. (Page 1, Line 41-44)
- P3: It should be mentioned earlier that GPC was done as well. The discussion can be as it is later on, but a brief mentioning would be good.
Response: Statement about GPC results is already present at Page 4, Line 154.
- P4: The characteristic of the PEG-grafted Geld should be pronounced better in the phrasing on this page, but also throughout the manuscript. It seems that the authors sometimes lose track of this important characteristic.
Response: The well-known characteristics of PEG have already been mentioned at Page 2, Line 85-88. Furthermore, PSA and PSA-g-mPEG12 networks have been clearly distinguished throughout the manuscript.
- P5-6: PEG45 melts higher that PEG23, but this is not reflected as such (only the higher degree of crystallinity is mentioned). The melting temperatures have to be cross-referenced with literature data.
Response: Statement about difference in melting temperature of PEG45 and PEG23 has been added with cross references (reference no. 44 and 45). (Page 6, Line 225-226).
- P6 (Table): Since PSA-g-mPEG12 allows for an absolute calculation of the molar mass, this should be used be used to back-calculate the original molar mass of PSA to check the value by GPC. GPC values can be off by a margin, so this rather important to check.
Response: The molar mass of graft copolymer (PSA-g-mPEG12) is calculated on the basis of the degree of grafting obtained from 1H NMR spectroscopy and taking into account molar mass of polymer (PSA) backbone obtained from GPC. Footnote has been changed to clarify that.
(Table 1, Page 6, Line 230-232)
- P7 - Fig. 5: The measurements appear to be done only once. They have to repeated to be triplicates in order to add error bars and put the results into perspective.
Response: Measurements were done in triplicate. Error bars (Fig. 5, Page 7) and text (Page 16, Line 617-618) have been added to the manuscript.
- P7 - Fig. 6: The figure caption says the gels swell in D2O while the text mentions H2 Please be consistent and correct where necessary.
Response: Since D2O was used, text was changed. (Page 6, Line 236 and Page 16, Line 612).
- P11 - Fig. 9: The figure actually contains error bars, but these are too small to be recognised (or are hard to spot). This should be stated in the figure caption.
Response: Symbols with error bars have been changed to be easily recognized. (Page 11, Fig. 9)